# Development of the Australian Dietary Guidelines Adherence Tool (ADG-AT): A Food Matching Protocol

**DOI:** 10.3390/nu17061071

**Published:** 2025-03-19

**Authors:** Rosa Piscioneri, Karen Zoszak, Yasmine Probst

**Affiliations:** School of Medical, Indigenous and Health Sciences, University of Wollongong, Wollongong, NSW 2522, Australiakaz914@uowmail.edu.au (K.Z.)

**Keywords:** food matching, Australian dietary guidelines, food groups, discretionary foods, adherence, chronic diseases, food frequency questionnaire, methodological

## Abstract

**Background/Objectives**: Food matching aligns food consumption and food composition data to quantify intakes of a food component or category. A systematic approach to food matching is required to obtain the highest quality match and, therefore, most accurately quantify the intake of the food component under investigation. This study aims to provide a tool to assess adherence with the Australian Dietary Guideline food group recommendations by the development of a food matching method that links dietary intake data from a food frequency questionnaire to food group data in the Australian Dietary Guideline database. **Methods**: Two researchers trained in food composition independently applied a stepwise approach to link the Dietary Questionnaire for Epidemiological Studies Version 2 food frequency questionnaire and the Australian Dietary Guideline database. Food preparation methods, mixed dishes and Australian Dietary Guideline database representative foods were considered to ensure the highest quality result. Average values were calculated for foods for which multiple items were matched. **Results**: The Australian Dietary Guideline Adherence Tool (ADG-AT) was produced, providing the number of servings of the five Australian Dietary Guideline food groups and discretionary foods per 100 g of food for 5742 food items. **Conclusions**: The ADG-AT produced in this study allows convenient evaluation of Australian Dietary Guideline adherence in studies using the Dietary Questionnaire for Epidemiological Studies food frequency questionnaire to collect dietary intake data. This informs the identification of dietary risk factors for nutritional inadequacy and chronic disease. The systematic methods used in this study can be reapplied to different dietary intake collection tools and food composition databases.

## 1. Introduction

National dietary guidelines aim to provide population-level dietary advice that is evidence-based, accessible and practical [1,2]. However, dietary recommendations are effective only when adhered to. In Australia, the 2022 National Healthy Survey (NHS) identified that only 4.2% of adults and 4.3% of children met the service recommendations for both fruits and vegetables [3]. Furthermore, discretionary foods (those high in saturated fat or containing added salt/sugar or alcohol) accounted for 35% of total energy intake [4]. Poor diet was estimated to contribute to 10% of Australian deaths in 2018, most of which were related to the highest cause of mortality worldwide—chronic disease [5,6]. Dietary risk factors that contribute to chronic diseases such as diabetes, hypertension and dyslipidaemia are global challenges contributed by factors such as a low intake of fruit, vegetables, milk, nuts, seeds and legumes, whole grains, dietary fibre, polyunsaturated fats, fish and seafood and a high intake of sodium, sugar-sweetened beverages and red or processed meats [7]. It is evident that populations are regularly engaging in dietary risks and the incidence of chronic disease is rising proportionately [7].

Dietary guidelines aim to ensure nutritional adequacy and prevent chronic disease [8]. Research indicates that greater adherence to dietary guidelines is associated with lower morbidity and mortality [9,10,11]. Long-term Australian Dietary Guideline (ADG, Table 1) adherence, for example, has been shown to be associated with a 7% reduction in all-cause and cardiovascular disease mortality in a 20-year cohort [9]. Adherence has also been associated with a reduced risk of type 2 diabetes, improved quality-of-life in older adults and a reduced risk of depressive symptoms in females [12]. Conversely, consumption of discretionary foods has been associated with insulin resistance, hyperlipaemia and hypertension [13,14]. This study focuses on Guidelines 2 and 3, namely, foods to consume daily and foods to limit [8].

Measuring adherence to dietary guidelines can be challenging. In the research setting, diet quality indices are often used to summarise a population’s dietary intake into a single measure of adherence to predefined criteria or guidelines [15]. Several of these are based on Australian guidelines, including the Australian Recommended Food Score-1 (ARFS-1) and the Dietary Guideline Index-2013 (DGI-2013) [15]. However, alternative approaches offer adherence insights further to diet quality alone.

Food matching is a method of pairing food consumption data with food composition data to quantify the intake of a particular food component, including nutrients, contaminants and additives [16]. This method is particularly suited to large-scale studies or agricultural research for financial evaluation, requiring a comprehensive food-matching protocol to accurately quantify the intake of the food component or category of concern [16]. The Food and Agriculture Organisation (FAO) acknowledges that while exact food matching is not always possible, it remains the most rigorous method of extrapolating food intake data into broader categories [16]. Food consumption data are collected using one of a variety of instruments (e.g., food frequency questionnaire (FFQ), 24 h recall (24HR)) and a food composition database that contains the food component under investigation and is appropriate for the study population. The matching process developed is, therefore, a unique, stepwise method, which can be reapplied in further studies using the same tools.

The ADG database was created to determine adherence with dietary guidelines by food group in the 2011-12 Australian Health Survey (AHS) [17,18]. The ADG database complements the Australian Food and Nutrient Database (AUSNUT 2011-13) by quantifying ‘serves/100 g’ for each food item reported in the AHS [19]. Unlike diet quality indices, the ADG database determines food group servings for individual food items and mixed dishes. The ADG database has been applied to 24 h recall dietary intake data in a clinical trial but not FFQ data in this setting [17,18]. The aim of this study was, therefore, to develop a systematic process to apply the ADG database to FFQ data using food matching techniques to enable researchers to determine adherence to food group recommendations in Australian adults. The result of this process is the Australian Dietary Guidelines Adherence Tool (ADG-AT).

## 2. Materials and Methods

This exploratory methodological study outlines the process of matching a dietary tool commonly used in Australian cohort studies with the dietary guidelines database. It outlines a systematic approach to developing a tool that can be used directly or adapted for use in other settings.

### 2.1. The Dietary Questionnaire for Epidemiological Studies Version 2

The Dietary Questionnaire for Epidemiological Studies Version 2 (DQES) was selected as the proxy FFQ in developing the ADG-AT as it is widely used among epidemiological studies in Australia. The DQES is a 74-item self-administered dietary assessment tool designed to assess habitual food and beverage intake in the 12 months prior to completion. The DQES was modified from an FFQ developed by Cancer Council Victoria in the 1980s to be suitable for use in the ethnically diverse Australian population [20,21]. The DQES was developed specifically for Australian adults [22] and has been validated for use in young adults aged 18–34 years [23] and women of child-bearing age [24]. Participants are prompted to report on food and beverage intake through a variety of semi-quantitative questions. Participants select how often they would typically consume each food from ‘Never’ to ‘3 or more times per day’ on a 10-point Likert scale. A majority of the FFQ categorises food items into four sections: cereal foods, sweets and snacks; dairy products, meat and fish; fruit; and vegetables (including fresh, frozen and tinned). These categories resemble the five food groups of the ADG, in addition to discretionary foods. Beverage intake is measured in a similar way, using juice, alcohol and water consumption probing questions. Portion size variation between participants is assessed using pictorial questions. All output data are calculated by the Cancer Epidemiology Centre at Cancer Council Victoria using Australian food composition data, resulting in 101 output food items available for analyses [21].

### 2.2. The Australian Dietary Guidelines Database

The ADG database was used to convert DQES food items from amount per day to the corresponding number of ADG food group servings. Both the DQES output fields and the ADG database were accessed in Microsoft Excel (Version 2021, Microsoft Corp, Redmond, WA, USA, 2021). The DQES output fields provide intake (g/day) of each food item (n = 101) included in the FFQ. In contrast, the ADG database specifies the number of ADG food group servings (servings/100 g) for each food item (n = 5742) using a food ID included in the 2011-12 AHS.

### 2.3. The Food Matching Method

All 101 food items in the DQES were systematically matched to one or more food items in the ADG database by two dietitian researchers (RP, KZ), independently. The principles of the data matching method were adapted from a previous study using diet history data and components of the FAO Guidelines for Food Matching [25].

Firstly, a string search of terms in the DQES food item description was undertaken in the ADG database (Figure 1). Further selection and exclusion criteria were then applied, resulting in one or more matches. Selection criteria included descriptions having search terms in the primary position (i.e., at the beginning of the description) and descriptions specifying “not further defined” (i.e., representing a summary of all other items of the same food). Exclusion criteria are shown in Table 2. Animal proteins and starchy vegetables including pulses were assumed to be cooked (as consumed) with food retention factors applied [26]. A new food item was created where multiple matches remained by calculating the mean value for all food groups. The food matching process and discretionary serve calculations are given in the Appendix A.

### 2.4. Identification and Quantification of Discretionary Foods

The ADGs describe discretionary foods as energy-dense and nutrient-poor, providing a discretionary flag to assist with the identification of these foods. The ADG Educator Guide states that one serving of discretionary food is equivalent to an amount of that food containing 600 kJ energy. DQES items were identified as a discretionary food if >20% of the ADG foods from which the average values were calculated were also classified as discretionary (Table 3). The number of ‘discretionary’ servings (distinct from food group servings) was calculated by dividing the total energy per 100 g by 600 kJ for each food. This approach is similar to a previous study that sought to clarify discretionary servings [14]. A worked example is given in the Appendix A.

## 3. Results

The product of the food matching process was a food group summary table that allows calculation of food group intake (total servings/day) for direct comparison with dietary guideline recommendations. As the aim of the ADG-AT was to assess dietary intake in adults, additional guidelines regarding a healthy weight, breastfeeding and safe food preparation were not included in this study’s working definition of ADG adherence. The final summary table showing servings/day for each food group and discretionary foods is provided in Table 4.

## 4. Discussion

In this study, we developed a systematic approach to matching items in a FFQ to align to the ADG database. Our methodological study was able to successfully match each of the DQES items to equivalent foods in the ADG database to create the ADG-AT. The tool provides the food group and discretionary food content per 100 g for each of the 101 food and beverage outputs related to the DQES questionnaire. This will enable researchers using the DQES FFQ to determine the number of ADG food group servings reported per day by participants, which can be compared with recommendations to evaluate adherence with dietary guidelines. As the DQES measures retrospective dietary intake over a 12-month period, the ADG-AT provides insight into usual ADG adherence for an individual and eating patterns within a population. The systematic process used to create the tool is applicable to other country guidelines and assessment tools.

The stepwise approach used in our study was undertaken previously by our team to create a matching file to convert the 2007 AUSNUT database to the 2011-13 version [27]. Changes in the food supply and commonly consumed foods necessitated the AUSNUT database update, meaning that studies that collected dietary data using the 2007 version needed updating as well [27]. The authors developed a comprehensive matching criteria that included matching generic and non-descript items; matching conceptually similar foods; and using professional judgement to determine when the nutrient composition of food items was similar [27]. While the present study matched different tools, similar concepts were implemented to strengthen the approach taken.

The ADG database has been applied previously: in the 2011-13 AHS and in a secondary analysis of a weight-loss intervention [13,28]. The latter matched food items reported by participants in a 24 h recall to the corresponding ADG database by Food ID to determine change in ADG adherence over a three-month period [13]. Our current study expanded this prior work by applying the ADG database to a FFQ, a dietary assessment data collection tool often used in observational studies. In the present study, the Food ID was not available for the DQES for direct matching; requiring a systematic and robust approach to food matching to be developed.

Dietary guideline ‘adherence’ has previously been inferred using diet quality indices with components that resemble the dietary guideline food groups [28]. Some of these indices have also been applied to the DQES FFQ; however, the approaches have limitations [29]. Firstly, a single diet quality score does not provide the food group-level. Secondly, the indices are often required to base conversion to dietary guideline servings on the examples provided in the dietary guidelines [30]. For example, in the dairy/alternatives food group, the only cheeses that the ADGs provides serve size examples for are hard and ricotta cheeses. The DQES as a dietary assessment tool provides more comprehensive data on cheese, including hard, firm, soft, cream and cottage cheeses [15]. When the ADG database is applied, this provides more specific and complete information regarding cheese consumption, resulting in more accurate data for reported dairy/alternatives consumption. While the example relates to Australian data, the principles are evident in other countries. For example, the Dietary Guidelines for Americans provide dairy serve sizes in terms of ‘1 cup equivalents’, equating to ‘1½ ounces natural cheese such as cheddar cheese’ or ‘2 ounces of processed cheese’ [31]. Clearly, these are broad categorisations intended for simplification of health messages; food composition analysis, however, can benefit from a greater degree of granularity. Further, ‘mixed’ (composite) dishes can also be problematic for dietary indices. The process used to create the ADG-AT enabled translation of these items to their multiple food group content. For example, while the DQES ‘hamburger’ is a discretionary food, it also contributes to ADG food groups (all except fruit). Composite food item reporting in dietary intake data is not unique to Australian data, but it is a global challenge that may impact the accuracy of reported outcomes. For example, a study by Bullock et al. found that there was wide variability in the scores of four diet quality indices when applied to composite meals [32]. ‘Dish composition’ databases have been applied to dish-based FFQs to address this challenge in Korean, Bangladeshi and Singaporean populations, in an approach similar to the ADG-AT [33].

The strengths of this study relate to the systematic approach taken to the development of the matching criteria, informed by prior research and undertaken independently by two researchers [13,27]. The same principles can be applied to adapt the ADG-AT to other tools for dietary intake data collection and food composition analysis. The ADG-AT is novel in that it enables ADG food group analysis of FFQ data for the first time. Furthermore, discretionary foods to limit are quantified rather than simply flagged as in the ADG database. This provides a more comprehensive evaluation of ADG adherence against the two guidelines relating to foods to consume daily and foods to consume on occasion [30].

Limitations of the study include a focus on the ADG food groups at the broadest level. Future studies could include other components of the ADGs, for example, wholegrains and refined grains and dairy products with higher, medium and lower levels of fat. Dietary variety could also be assessed to provide a more complete evaluation of adherence with Guideline 2. Secondly, the food group contributions of the multiple ADG foods included in each DQES item would ideally be weighted according to consumption patterns rather than an average taken. This was not possible due to the unavailability of reliable consumption data for the population to which the summary table was applied. Finally, the authors acknowledge that the DQES v2 is an outdated tool that has been superseded, however, given the abundance of data collected using this version including ongoing active studies, its selection is justified. The ADG-AT would need to be updated to reflect later versions of the DQES.

## 5. Conclusions

This study created the ADG-T that was developed following a systematic food matching method. The tool enables the ADG database to be applied to the DQES FFQ. The ADG-T will enable studies using the DQES FFQ to determine daily food group reporting based on the dietary guidelines, including reported discretionary food consumption, and can be adapted for use by other countries, thereby allowing the identification of dietary risk factors for nutritional inadequacy and chronic disease.

## Figures and Tables

**Figure 1 nutrients-17-01071-f001:**
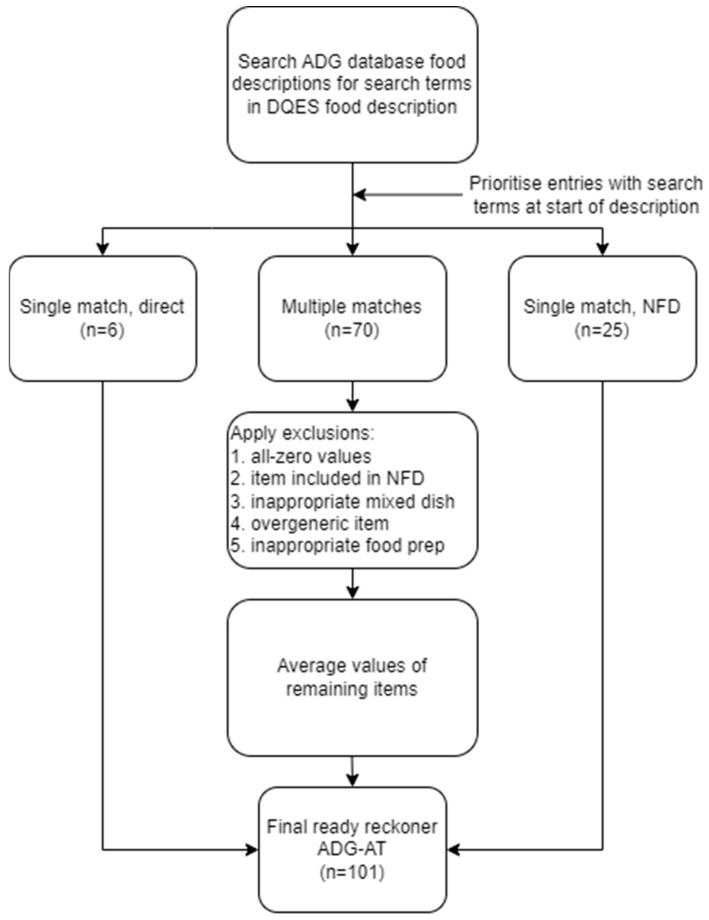
Food matching process flow: matching Dietary Questionnaire for Epidemiological Studies Version 2 (DQES) food items to Australian Dietary Guidelines database (ADG database) entries. NFD = “not further defined”, a representative summary item for similar entries.

**Table 1 nutrients-17-01071-t001:** Summary of the Australian Dietary Guidelines 2013.

Guideline 1	To achieve and maintain a healthy weight, be physically active and choose amounts of nutritious food and drinks to meet your energy needs.
Guideline 2	Enjoy a wide variety of nutritious foods from these five food groups every day: Vegetables and legumes/beans; Fruit; Grain (cereal) foods, mostly wholegrain and/or high fibre; Lean meats and poultry, fish, eggs, tofu, nuts, seeds, legumes/beans; Milk, yoghurt, cheese and/or alternatives, mostly reduced fat.
Guideline 3	Limit intake of foods containing saturated fat, added salt, added sugars and alcohol.
Guideline 4	Encourage support and promote breastfeeding.
Guideline 5	Care for your food; prepare and store it safely.

**Table 2 nutrients-17-01071-t002:** Exclusion criteria for matching DQES food items to ADG database entries.

Exclusion Criteria	DQES Example
Food items with all values equal to zero	“Beef, all cuts, separable fat, grilled or roasted without fat”
Food items not in their ‘typically consumed’ form	“Chicken, breast, flesh, raw”
Food items in mixed dishes where a single food is implied by DQES e.g., “Capsicum”	“Capsicum, stuffed with meat & rice”
Food items with general descriptions when the DQES description is specific e.g., “Wholemeal bread”	“Bread, commercial, fresh, not further defined”
Inappropriate food form	Raw animal proteins and starchy vegetables

**Table 3 nutrients-17-01071-t003:** DQES items identified as discretionary foods, several spanning multiple classifications.

Recommendation	DQES Foods
Limit saturated fat	Bacon, biscuits, butter, cakes, chips, chocolate, crisps, deli meats, hamburgers, ice cream, pies, pizza, sausages
Limit added salt	Bacon, chips, crisps, deli meats, pies, pizza, sausages, Vegemite
Limit added sugar	Biscuits, cakes/pastries, chocolate, ice cream, jam, sugar
Limit alcohol	Beer, fortified wine, liqueurs, spirits, wine

**Table 4 nutrients-17-01071-t004:** Australian Dietary Guidelines Adherence Tool (ADG-AT) food group summary table, providing number of servings of the five food groups and discretionary foods per DQES food item.

				Servings/100 g				
DQES Food Name	Form	Method	Grains	Vegetables/Legumes	Fruit	Dairy/ Alternatives	Meat/ Alternatives	Discretionary
All Bran	processed	not further defined	3.3	0.0	0.0	0.0	0.0	0.0
Apples	raw	not further defined	0.0	0.0	0.7	0.0	0.0	0.0
Apricots	raw	single	0.0	0.0	0.7	0.0	0.0	0.0
Avocado	raw	average	0.0	1.3	0.0	0.0	0.0	0.0
Bacon	cooked	average	0.0	0.0	0.0	0.0	1.4	1.9
Baked beans	processed	not further defined	0.0	1.3	0.0	0.0	0.7	0.0
Bananas	raw	average	0.0	0.0	0.7	0.0	0.0	0.0
Bean sprouts, alfalfa sprouts	raw/cooked	average	0.0	1.4	0.0	0.0	0.0	0.0
Beef	cooked	average	0.0	0.0	0.0	0.0	1.5	0.0
Beer—full strength	processed	average	0.0	0.0	0.0	0.0	0.0	0.3
Beer—low alcohol	processed	average	0.0	0.0	0.0	0.0	0.0	0.2
Beetroot	raw/cooked	average	0.0	1.2	0.0	0.0	0.0	0.0
Bread—high fibre white	processed	average	2.7	0.0	0.0	0.0	0.0	0.0
Bread—multi-grain	processed	average	2.7	0.0	0.0	0.0	0.0	0.0
Bread—rye	processed	average	2.7	0.0	0.0	0.0	0.0	0.0
Bread—white	processed	average	2.7	0.0	0.0	0.0	0.0	0.0
Bread—wholemeal	processed	average	2.7	0.0	0.0	0.0	0.0	0.0
Broccoli	raw/cooked	average	0.0	1.3	0.0	0.0	0.0	0.0
Butter	processed	not further defined	0.0	0.0	0.0	0.0	0.0	5.0
Cabbage, Brussels sprouts	raw/cooked	average	0.0	1.3	0.0	0.0	0.0	0.0
Cakes, sweet pies, tarts, other sweet pastries	cooked	average	0.9	0.0	0.2	0.1	0.1	2.3
Carrots	raw/cooked	average	0.0	1.3	0.0	0.0	0.0	0.0
Cauliflower	raw/cooked	average	0.0	1.2	0.0	0.0	0.0	0.0
Celery	raw/cooked	average	0.0	1.2	0.0	0.0	0.0	0.0
Cheese—cream	processed	not further defined	0.0	0.0	0.0	2.5	0.0	0.0
Cheese—firm (cheddar, edam)	processed	average	0.0	0.0	0.0	2.5	0.0	0.0
Cheese—hard (parmesan, romano)	processed	average	0.0	0.0	0.0	2.5	0.0	0.0
Cheese—low fat	processed	single	0.0	0.0	0.0	2.5	0.0	0.0
Cheese—ricotta, cottage	processed	average	0.0	0.0	0.0	0.8	0.0	0.0
Cheese—soft (camembert, brie)	processed	not further defined	0.0	0.0	0.0	2.5	0.0	0.0
Chicken	cooked	average	0.2	0.0	0.0	0.0	1.1	0.0
Chocolate	processed	not further defined	0.0	0.0	0.0	0.8	0.0	3.5
Corn chips, potato crisps, Twisties	processed	average	2.2	0.7	0.0	0.1	0.0	3.5
Corned beef, luncheon meats, salami	processed	average	0.0	0.0	0.0	0.0	1.4	1.5
Cornflakes, Nutrigrain, Special K	processed	average	3.3	0.0	0.0	0.0	0.0	0.0
Crackers, crispbreads, dry biscuits	processed	not further defined	2.9	0.0	0.0	0.0	0.0	0.0
Cucumber	raw/cooked	average	0.0	1.3	0.0	0.0	0.0	0.0
Eggs	cooked	average	0.0	0.0	0.0	0.0	0.8	0.0
Fish—fried (including take-away)	cooked	average	0.0	0.0	0.0	0.0	1.0	0.0
Fish—steamed, grilled, baked	cooked	average	0.0	0.0	0.0	0.0	1.0	0.0
Fish—tinned (salmon, tuna, sardines)	processed	average	0.0	0.0	0.0	0.0	0.9	0.0
Flavoured milk drink (cocoa, Milo)	processed	average	0.0	0.0	0.0	0.3	0.0	0.0
Fortified wines (port, sherry)	processed	average	0.0	0.0	0.0	0.0	0.0	0.9
Fruit—tinned, frozen	processed	average	0.0	0.0	0.6	0.0	0.0	0.0
Fruit juice	processed	not further defined	0.0	0.0	0.8	0.0	0.0	0.0
Garlic	raw/cooked	average	0.0	1.5	0.0	0.0	0.0	0.0
Green beans	raw/cooked	average	0.0	1.3	0.0	0.0	0.0	0.0
Ham	processed	average	0.0	0.0	0.0	0.0	1.5	0.8
Hamburger with bun	mixed	average	0.9	0.2	0.0	0.2	0.5	1.7
Ice cream	processed	not further defined	0.0	0.0	0.0	0.5	0.0	1.3
Jam, marmalade, honey, syrups	processed	average	0.0	0.0	0.2	0.0	0.0	1.6
Lamb	cooked	average	0.0	0.0	0.0	0.0	1.5	0.0
Lettuce, endive, other salad greens	raw/cooked	average	0.0	1.5	0.0	0.0	0.0	0.0
Mango, paw paw	raw	single	0.0	0.0	0.7	0.0	0.0	0.0
Meat pies, pasties, quiche, other savoury pastries	cooked	average	0.7	0.3	0.0	0.1	0.3	1.7
Milk—full cream	processed	not further defined	0.0	0.0	0.0	0.4	0.0	0.0
Milk—reduced fat	processed	not further defined	0.0	0.0	0.0	0.4	0.0	0.0
Milk—skim	processed	not further defined	0.0	0.0	0.0	0.4	0.0	0.0
Milk—soya	processed	not further defined	0.0	0.0	0.0	0.4	0.0	0.0
Muesli	processed	average	2.3	0.0	0.5	0.0	0.4	0.0
Mushrooms	raw/cooked	average	0.0	1.2	0.0	0.0	0.0	0.0
Nuts	processed	average	0.0	0.0	0.0	0.0	3.3	0.0
Onion, leeks	raw/cooked	average	0.0	1.1	0.0	0.0	0.0	0.0
Oranges, other citrus	raw	average	0.0	0.0	0.7	0.0	0.0	0.0
Other beans (chick peas, lentils)	cooked	average	0.0	1.3	0.0	0.0	0.6	0.0
Pasta or noodles (including lasagne)	cooked	average	1.3	0.0	0.0	0.0	0.0	0.0
Peaches, nectarines	raw	average	0.0	0.0	0.7	0.0	0.0	0.0
Peanut butter, peanut paste	processed	not further defined	0.0	0.0	0.0	0.0	2.9	0.0
Pears	Raw	not further defined	0.0	0.0	0.7	0.0	0.0	0.0
Peas	cooked	average	0.0	1.3	0.0	0.0	0.1	0.0
Peppers (capsicum)	raw/cooked	average	0.0	1.3	0.0	0.0	0.0	0.0
Pineapple	raw	single	0.0	0.0	0.7	0.0	0.0	0.0
Pizza	cooked	average	1.4	0.3	0.0	0.5	0.2	1.8
Pork	cooked	average	0.0	0.0	0.0	0.0	1.5	0.0
Porridge	cooked	average	0.8	0.0	0.0	0.3	0.0	0.0
Potatoes—cooked without fat	cooked	average	0.0	1.3	0.0	0.0	0.0	0.0
Potatoes—roasted, fried (including hot chips)	cooked	average	0.1	1.3	0.0	0.0	0.0	1.6
Pumpkin	cooked	average	0.0	1.3	0.0	0.0	0.0	0.0
Red wine	processed	single	0.0	0.0	0.0	0.0	0.0	0.5
Rice	cooked	average	1.4	0.0	0.0	0.0	0.0	0.0
Sausages, frankfurters	cooked	average	0.0	0.0	0.0	0.0	1.2	1.6
Silverbeet, spinach	raw/cooked	average	0.0	1.3	0.0	0.0	0.0	0.0
Soy beans, soy bean curd, tofu	processed	average	0.0	0.3	0.0	0.0	0.5	0.0
Spirits, liqueurs	processed	average	0.0	0.0	0.0	0.1	0.0	1.8
Spread—butter and margarine blend	processed	not further defined	0.0	0.0	0.0	0.0	0.0	4.4
Spread—margarine	processed	average	0.0	0.0	0.0	0.0	0.0	0.0
Spread—monounsaturated margarine	processed	not further defined	0.0	0.0	0.0	0.0	0.0	0.0
Spread—polyunsaturated margarine	processed	not further defined	0.0	0.0	0.0	0.0	0.0	0.0
Strawberries	raw	single	0.0	0.0	0.7	0.0	0.0	0.0
Sugar	processed	average	0.0	0.0	0.0	0.0	0.0	2.7
Sultana Bran, FibrePlus, Branflakes	processed	average	2.6	0.0	0.3	0.0	0.1	0.0
Sweet biscuits	processed	not further defined	1.6	0.0	0.0	0.1	0.0	3.3
Tomato sauce, tomato paste, dried tomatoes	processed	average	0.0	2.3	0.0	0.0	0.0	0.0
Tomatoes—fresh, tinned	raw/cooked	average	0.0	1.4	0.0	0.0	0.0	0.0
Veal	cooked	average	0.1	0.0	0.0	0.0	1.5	0.0
Vegemite, Marmite, Promite	processed	not further defined	0.0	0.0	0.0	0.0	0.0	1.1
Watermelon, rockmelon (cantaloupe), honeydew	raw	not further defined	0.0	0.0	0.7	0.0	0.0	0.0
Weet Bix, Vita Brits, Weeties	processed	not further defined	3.3	0.0	0.0	0.0	0.0	0.0
White wine (including sparkling)	processed	not further defined	0.0	0.0	0.0	0.0	0.0	0.5
Yoghurt	processed	not further defined	0.0	0.0	0.0	0.5	0.0	0.0
Zucchini	raw/cooked	average	0.0	1.2	0.0	0.0	0.0	0.0

## Data Availability

The Australian Dietary Guidelines database is available for download here: https://www.foodstandards.gov.au/science-data/monitoringnutrients/ahs-aus-dietary-guidelines (accessed on 16 March 2025). The discretionary food list is available for download here: https://www.abs.gov.au/AUSSTATS/abs@.nsf/DetailsPage/4363.0.55.0012011-13?OpenDocument (accessed on 16 March 2025).

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
