# Peer review of "Development of the Australian Dietary Guidelines Adherence Tool (ADG-AT): A Food Matching Protocol"

_nutrients, 2025, doi:10.3390/nu17061071_

Round 1
Reviewer 1 Report
Comments and Suggestions for Authors
In the abstract, it is not clear the results obtained in this study. The authors should present these results clearly and in a quantitative way.
Abbreviations should be avoided as keywords.
Regarding the topics covered in this manuscript, what is the outlook at a global level and not just in Australia? Improve the introductory section taking this in mind.
The identification and quantification of discretionary foods must be better explained (section 2.4).
The Results section should be separated from Material and Methods.
The discussion of the obtained results should be made taking into consideration a worldwide perspective. Include more studies from other countries to improve the analysis of your research in this section. Conclusions should also be improved considering this.
Author Response
Thank you for your review. Please refer to the attachment.

Reviewer 2 Report
Comments and Suggestions for Authors
Many thanks to the editor for the opportunity to revise the manuscript.
In this study, Piscioneri et al. aim to develop a tool that, by matching the items of a validated FFQ for the Australian population with the Australian Dietary Guidelines Database, can provide a daily consumption portion for each item, potentially facilitating the assessment of adherence to national nutritional guidelines in large-scale studies. Below are my comments and suggestions.
It would be better to use the expression “assess adherence to the Australian dietary guidelines” and “assess habitual food” rather than “measure” (line 10 and 94). The term “measure” is more appropriate when referring to an instrumental outcome, rather than an outcome such as adherence or dietary habits.
The information regarding the FFQ used is not very clear. The authors state that the FFQ they considered (the Dietary Questionnaire for Epidemiological Studies Version 2 (DQES v2)) consists of 101 items. However, the references provided contradict this information. Reference number 20 indicates: 'The DQES v2 comprises a food list of 74 items, grouped into four food type categories.' Reference number 21 refers to 'The Melbourne FFQ' and not the DQES v2 considered in this study. Reference number 22 refers to 'The Dietary Questionnaire for Epidemiological Studies Version 3.2 (DQES v3.2),' which is described as 'an update of the widely used DQES v2 composed of 80 items within five types of dietary intake.' Reference number 24 refers to 'the Anti Cancer Council of Victoria food frequency questionnaire (ACCVFFQ) relative to women of child-bearing age.' Also the information regarding age is not clear: plausibly, it might be more accurate to consider validity for the 18-34 and 40-69 age ranges, since reference number 23 refers to the validation of the FFQ in the 18-34 age range, while reference number 20 (regarding the DQES v2) indicates the 40-69 age range for the sample in which the FFQ was validated. I suggest that the authors provide clarification on these aspects.
In the matching process described in paragraph 2.3, the authors state that they assumed protein foods from animal sources and starchy vegetables were considered cooked rather than raw. Could an explanation be provided for this assumption, including the cooking method considered and whether any food conversion factors from raw to cooked were applied (as referenced in reference number 25)? Additionally, I believe the sentence on lines 127-128 might be somewhat misleading. Upon reading it, it seems that legumes are classified as starchy vegetables (they are an important source of plant-based proteins, but they are not considered vegetables).
Author Response

(The authors gave the same response as above.)
